# Inflated Ovary May Increase the Dispersal Ability of Three Species in the Cold Deserts of Central Asia

**DOI:** 10.3390/plants12101950

**Published:** 2023-05-10

**Authors:** Jannathan Mamut, Kewei Chen, Carol C. Baskin, Dunyan Tan

**Affiliations:** 1College of Life Science, Xinjiang Agricultural University, Ürümqi 830052, China; jinaiti@163.com (J.M.); ckw1157@163.com (K.C.); carol.baskin@uky.edu (C.C.B.); 2Key Laboratory of Ministry of Education for Western Arid Region Grassland Resources and Ecology, College of Grassland Sciences, Xinjiang Agricultural University, Ürümqi 830052, China; 3Department of Biology, University of Kentucky, Lexington, KY 40506, USA; 4Department of Plant and Soil Sciences, University of Kentucky, Lexington, KY 40546, USA

**Keywords:** cold desert, inflated ovary, morphological traits, seed dispersal

## Abstract

Among the diaspores of angiosperms an inflated ovary (IO) is a novel morphological trait, but no studies have evaluated its effects on dispersal. The primary aim of this study was to determine the effect of the IO on diaspore dispersal in three cold desert species (*Carex physodes*, *Calligonum junceum*, and *Sphaerophysa salsula*). Various morphological features and the mass of fruits and seeds of each species were measured. The role of an IO in diaspore dispersal by wind and water was determined by comparing responses of intact (inflated) IOs and flattened fruits and seeds. Mature diaspores of three species were dispersed by wind, and the IO significantly increased dispersal distance in the field and at different wind speeds in the laboratory. The floating time on water was greater for inflated fruits than flattened fruits and seeds. Since the seed remains inside the IO until after dispersal is completed, the IO of the three species enhances diaspore dispersal. This is the first detailed study on how an IO increases diaspore/seed dispersal. Furthermore, after primary dispersal by wind, secondary dispersal can occur via wind or surface runoff of water, and each method is enhanced by the presence of an IO.

## 1. Introduction

Dispersal is a key stage in the life history of plants, and it is an important ecological process that affects the seed fate, spatial distribution and dynamics of plant species, the structure of communities and the establishment of ecosystem function [1,2,3,4]. The dispersal of diaspores (seeds, fruits, infructescences, and other dispersal units) can be achieved by animals [5], wind [6], or water [7], and plant diaspore traits are directly related to their habitats and dispersal strategies [8,9,10,11,12]. Furthermore, the dispersion-related traits of diaspores affect dispersal patterns [13,14,15,16], and they may vary in mass and size or have different dispersal appendages [17,18,19].

Angiosperms have evolved a diversity of morphological structures, including hairs [20], wings [14,21], awns [22], spines [23,24], bracts [25,26], elaiosomes [27,28], mucilage [29,30], and inflated calyx [31], which improve the diaspore dispersal capacity via different vectors [32]. Calyx inflation occurs during fruit maturation and results in a hollow, gas-filled structure [33,34], which has primarily been observed in taxa such as *Anisodus*, *Nicandra*, *Physaliastrum*, *Physalis*, *Physochlaina*, *Przewalskia*, and *Withania* in the Solanaceae, *Coleus* in the Labiatae, and *Astragalus* in the Fabaceae [33,35,36]. In addition, we have observed that the ovary is inflated in species of *Astragalus*, *Oxytropis*, *Glycyrrhiza*, and *Halimodendron* in the Fabaceae and *Leontice* in Berberidaceae in the cold deserts of Central Asia.

In some species, an inflated calyx improves fruit and seed dispersal by wind or water, but the role of an inflated ovary (IO) in fruit and seed dispersal has not been investigated [31,36,37]. The primary purpose of this study was to determine the effect of the IO on the seed dispersal of the cold desert species *Carex physodes* M.Bieb. (Cyperaceae) (Figure 1A,B), *Calligonum junceum* (Fisch. & C.A.Mey.) (Polygonaceae) (Figure 1F,G), and *Sphaerophysa salsula* (Pall.) DC. (Fabaceae) (Figure 1K,L). These species are distributed in the Junggar Basin of Xinjiang Uyghur Autonomous Region in northwest China. We hypothesized that the presence of an IO significantly enhances the diaspore/seed dispersal of the three species in their cold desert habitat. To test the hypothesis, we compared the dispersal ability of intact (inflated IOs) and flatten fruits and isolated seeds.

## 2. Results

### 2.1. Morphological Characteristics of Fruits and Seeds

The mass, length, width, and thickness of fruits and seeds for each species are presented in Table 1. Fruits of *C*. *physodes* were achenes, and they were oval or subcircular, biconvex, and light brown at maturity (Figure 1C); the distance from the seed to the perigynium was 11.19 ± 0.15 mm (Figure 1D). Each *C*. *physodes* fruit contained one seed, which was oval or subcircular and pale yellow (Figure 1E). Fruits of *C*. *junceum* were achenes, and they were round or broadly ellipsoidal, and reddish-brown at maturity (Figure 1H); the distance from the seed to the pericarp was 3.28 ± 0.06 mm (Figure 1I). Each *C*. *junceum* fruit contained one seed, which was quadrangular, stellate in cross-section and yellowish (Figure 1J). Fruits of *S*. *salsula* were pods, and they were ellipsoid or ovoid, and yellowish-white at maturity (Figure 1M); the distance from seed to pericarp was 6.58 ± 0.20 mm (Figure 1N). Each *S*. *salsula* fruit contained 35 ± 1 seeds, which were reniform or subsemi-orbicular and brown in color (Figure 1O).

### 2.2. Natural Dispersal of Inflated Ovaries

In the natural habitat, dispersal (by wind) of the IOs of *C*. *physodes* and *C*. *junceum* began at fruit maturity in mid-May and was completed by mid-August. The dispersal of the IOs of *S*. *salsula* began at fruit maturity in mid-June and was completed by mid-July. The dispersal distance of inflated fruits was significantly greater than that of flattened fruits for all three species (*C*. *physodes*: *F* = 54.277, *p* < 0.001; *C*. *junceum*: *F* = 177.454, *p* < 0.001; *S*. *salsula*: *F* = 68.008, *p* < 0.001) (Figure 2).

### 2.3. Effect of an Inflated Ovary on Dispersal by Wind

In the laboratory, the time of descent in still air for *C*. *physodes* was inflated = flattened > seeds (*F* = 152.749, *p* < 0.001), but for *C*. *junceum* (*H* = 84.493, *p* < 0.001) and *S*. *salsula* (*F* = 210.891, *p* < 0.001) it was inflated fruits > flattened fruits > seeds (Figure 3A). Compared to seeds, inflated fruits descended at the slowest rate (*C*. *physodes*: *F* = 152.692, *p* < 0.001; *C*. *junceum*: *H* = 84.493, *p* < 0.001; *S*. *salsula*: *H* = 104.745, *p* < 0.001) (Figure 3B).

At a wind speed of 4 m·s^−1^, the dispersal distance of inflated and flattened *S*. *salsula* fruits did not differ (*F* = 66.344, *p* < 0.001) but the dispersal distance of *C*. *physodes* (*F* = 251.639, *p* < 0.001) and *C*. *junceum* (*H* = 100.791, *p* < 0.001) was inflated fruits > flattened fruits > seeds (Figure 3C). At a wind speed of 6 m·s^−1^, the dispersal distance was inflated fruits > flattened fruits > seeds (*C*. *physodes*: *F* = 89.684, *p* < 0.001; *C*. *junceum*: *H* = 123.193, *p* < 0.001; *S*. *salsula*: *H* = 82.702, *p* < 0.001) (Figure 3D).

### 2.4. Effect of an Inflated Ovary on Dispersal by Water

The fruits and seeds of the three species floated differently in still water. At 11 days, all seeds of *C*. *physodes* had dropped from the water surface, whereas 88% and 64% of the inflated and flattened fruits, respectively, were floating (Figure 4A). All seeds and flattened fruits of *C*. *junceum* had dropped from the water surface by 7 days; however, 80% of the inflated fruits were floating at 11 days (Figure 4B). At 10 days, all seeds of *S*. *salsula* had dropped, whereas 90% of the inflated fruits and 6% of the flattened fruits were floating (Figure 4C).

At a shaking speed of 80 rpm, all seeds of the three species had dropped from the water surface within 10 min (Figure 4D–F), but after 3 h at this speed 100% of the inflated fruits of *C*. *physodes* and *C*. *junceum* and 66% of the inflated fruits of *S*. *salsula* were floating. The final floating percentage of inflated fruits was significantly higher than that of seeds (*p* < 0.001) (Figure 4D–F). At a shaking speed of 160 rpm, all seeds of the three species dropped within 10 min (Figure 4G–I). After 3 h at 160 rpm, all the inflated fruits of *C*. *physodes* and *C*. *junceum*, 40% of the inflated fruits of *S*. *salsula*, 46% of the flattened fruits of *C*. *junceum* and <5% of the flattened fruits of *S*. *salsula* were still floating. The final floating percentage of inflated fruits was significantly higher than that of seeds (*p* < 0.001) (Figure 4G–I).

### 2.5. Correlation between Morphological and Wind Dispersal Traits of Inflated Ovary

The falling (descent) velocity was positively correlated with fruit mass (*r* = 0.32, *p* < 0.001), fruit width (*r* = 0.37, *p* < 0.001), and fruit thickness (*r* = 0.46, *p* < 0.001). The horizontal dispersal distance was negatively correlated with fruit mass (*r* = −0.63, *p* < 0.001), fruit width (*r* = −0.67, *p* < 0.001), and fruit thickness (*r* = −0.73, *p* < 0.001). The buoyancy exponent was negatively correlated with fruit mass (*r* = −0.51, *p* < 0.001), fruit width (*r* = −0.50, *p* < 0.001), fruit thickness (*r* = −0.62, *p* < 0.001), and descent velocity (*r* = −0.64, *p* < 0.001), but it was positively correlated with horizontal dispersal distance (*r* = 0.62, *p* < 0.001) (Figure 5).

## 3. Discussion

Our studies showed that the presence of an IO significantly increased the dispersal distance/time of fruits/seeds of *C*. *physodes*, *C*. *junceum*, and *S*. *salsula* by wind and water. Thus, our hypothesis that an IO would enhance dispersal was supported.

Generally, the fruits of *C*. *physodes*, *C*. *junceum*, and *S*. *salsula* were released from the mother plant by wind from mid-May to mid-August, when winds are most prevalent in the Junggar Basin [38], which supports the conclusion of Truscott et al. [39] that the dispersal success of diaspores is usually related to the weather conditions during the dispersal period. Similar to our results, an inflated calyx significantly increased the dispersal distance of *Physalis floridana* by wind [31].

The descent velocity of the diaspores in still air indicates how long it takes them to fall, and with an increase in falling time it is assumed there is an increase in dispersal distance [40]. That is, the longer the diaspores float in the air, the slower the descent velocity, and the greater the dispersal distance [41]. Thus, for wind-dispersed diaspores, the descent velocity is an important factor in determining how far from the mother plant diaspores will travel before they fall to the soil surface [42]. In our study, the descent velocity of inflated fruits was much lower than that of flattened fruits and seeds. The presence or absence of inflated fruits had a substantial effect on the dispersal distance at different wind speeds, and inflated fruits were dispersed a greater distance than flattened fruits and seeds at wind speeds of both 4 m·s^−1^ and 6 m·s^−1^ (Figure 3). The dispersal distances of the three species increased with increasing wind speed in our study, with the low mass fruits of *C. physodes* showing the greatest increase in dispersal distance (Table 1, Figure 3). Not surprisingly, the mass of the diaspores was negatively correlated with their dispersal distance [43,44], and the greater the mass of the diaspores, the shorter their horizontal dispersal distance [45,46]. Our results showed a negative correlation between the horizontal dispersal distance of diaspores and the descent velocity (Figure 5), which was similar to earlier research on fruit wings [47,48].

It has been shown that wind-dispersed diaspores also can be successfully dispersed by water [49]. Our water-dispersal experiments with different treatments of the three species revealed that inflated fruits could float for a longer period of time in still water than flattened fruits and seeds; this also was true for shaking water (Figure 4). Thus, there was a positive correlation between horizontal dispersal distance via wind and the buoyancy in water; similar results were obtained for *Brandisia hancei* seeds, which have wings. Diaspore wings increased the dispersal distance by wind due to increased seed buoyancy [50,51,52] and enhanced hydrochory via surface runoff [53].

The secondary dispersal of diaspores is one of the important mechanisms for dispersal desert plants [54,55]. The movement patterns of diaspores on the ground, e.g., turning, sliding or spinning, determine their secondary dispersal distance [56]. In our study, the diaspore shapes of three studied species were either elliptical or spherical. To our knowledge, this is the first detailed study on how an IO may increase diaspore/seed dispersal. Furthermore, previous studies have shown that in the wild, the dispersal ability of diaspores is independent of their germination [57]. That is, seed dispersal distance does not directly provide information on germination success. Overall, our studies showed that an IO may enable most seeds to be dispersed into a depression far from the mother plant by either wind or water, i.e., increasing seed distribution to a suitable environment for germination and seedling establishment in the Junggar Desert. However, further studies are required to address the germination success of the dispersed seeds. Further research is needed to investigate whether the prolonged soaking of fruits in water affects their germination.

In conclusion, the primary dispersal manner of the inflated fruits of *C*. *physodes*, *C*. *junceum*, and *S*. *salsula* is by wind, but secondary dispersal, or movement away from their initial landing location to a new environment, maybe more important than the primary dispersal by wind [9,58]. In the case of the three species with an IO, secondary dispersal could be via wind or surface run-off of water, and in either case the IO would enhance dispersal.

## 4. Materials and Methods

### 4.1. Diaspore Collection and Field Site Description

Freshly matured diaspores were collected from many individuals of the three species growing in natural populations in the Karamaili Mountains Nature Reserve (88°30′–90°03′ E, 44°36′–46°00′ N, altitude 600–1464 m) (Figure 6) in the northeastern part of the Junggar Basin, Xinjiang Uyghur Autonomous Region, China, from May to July 2021. Diaspores were stored in paper bags under ambient conditions in the laboratory (room temperature 28–30 °C and relative humidity 20–30%) for 1–3 months.

This cold desert has gravelly desert soil and a temperate, continental, arid climate. Mean annual temperature is 3.8 °C, and mean temperature of the coldest (January) and hottest (July) months is −19.3 °C and 22.8 °C, respectively. Average annual precipitation (including rain and snow) is 209.6 mm (National Meteorological Information Center, China Meteorological Administration), and annual potential evaporation is >2000 mm [59].

### 4.2. Morphological Characteristics of Fruits and Seeds

Color, shape, size, and mass of fruits and seeds of each species were observed/measured after they had been stored in the laboratory for 2 weeks. Length, width and thickness of fruits and seeds, and distance between the seed and the pericarp, were measured using 111N-102-10G digital calipers (0.01 mm); 50 measurements were made for each characteristic for each species. To determine mass, 10 replications of 100 fruits or seeds of each species were weighed using a Sartorius BS210S electronic balance (0.0001 g).

### 4.3. Natural Dispersal of Inflated Ovaries

To test the role of the IO on fruit/seed dispersal in the natural habitat, we selected 60 plants of *C*. *physodes*, *C*. *junceum*, and *S*. *salsula* on 7 May, 1 June, and 27 June 2021, respectively. All fruits on each plant were assigned to one of the following treatments: (1) naturally inflated fruits and (2) flattened fruits. To flatten the fruits, the air inside the fruit was expelled manually, and the fruit was held flat by clear tape around the outside. All fruits in the two treatments were sprayed with different colors of aerosol paint just before the fruits matured. The dispersal time and distance from the mother plants were recorded during weekly visits to the field sites.

### 4.4. Effect of Inflated Ovaries on Dispersal by Wind

On 5 September 2021, the dispersal of (1) naturally inflated fruits, (2) flattened fruits and (3) seeds of the three species was compared in the laboratory using the following procedures.

(a)Dispersal in still air. To measure fall rate in still air, a tube 200 cm tall and 15 cm in diameter was used. For each species, 100 inflated fruits, 100 flattened fruits and 100 seeds were released individually from the top of the tube, and the time required for them to fall from the release point to the bottom of the tube was measured with a digital stopwatch. The fall rate (cm·s^−1^) was calculated over this height [60,61].(b)Dispersal in flowing air. For each species, 100 inflated fruits, 100 flattened fruits and 100 seeds were dropped from the mean plant height of each species in the wild (20 cm for *C*. *physodes*, 70 cm for *C*. *junceum*, and 45 cm for *S*. *salsula*). Fruits/seeds were released one by one above a point on the floor and exposed to wind speeds of 4 or 6 m·s^−1^ generated by using different settings on an electric fan, and the distance diaspores traveled was measured to 0.001 m [62]. An anemometer was used to measure the wind speed at the release point.

### 4.5. Effect of Inflated Ovaries on Dispersal by Water

In the natural condition, the surface runoff of water following a precipitation event may have an impact on fruit/seed dispersal over the soil surface. To test the effects of an IO on dispersal potential by surface water runoff, the water-floating traits of fruits and seeds of each species were compared. From 5 to 17 September 2021, an experiment for the three species was conducted in a windless laboratory (room temperature 28–30 °C and relative humidity 20–30%) and involved three treatments using the following procedures.

(a)Floating in still water. To test buoyancy, 10 replicates of 10 inflated fruits, 10 flattened fruits and 10 seeds were used, and each replicate was placed in a plastic container, which was 10 cm in diameter and 20 cm deep and filled with 500 mL distilled water [63] [3 diaspore/seed treatments × 10 replicates = 30 containers]. The number of floating diaspores was recorded after 1, 2, 4, 6, and 8 h, and then after every 12 h for 11 days.(b)Floating in shaking water. Floating of intact fruits, flattened fruits and seeds was determined in disturbed water, i.e., at different shaker speeds. Fruits and seeds were placed 60 plastic containers of water, as described above [3 diaspore/seed treatments × two shaker speeds × 10 replicates = 60 containers]. The containers were placed on a shaker (model ST-98A, Shinetek Instruments, Beijing, China), and two shaker speeds of 80 and 160 revolutions per minute (rpm) were used to reduce the influence of surface tension and to simulate water movement [39,64]. Diaspores in 30 containers each were exposed to each speed for 180 min. The number of floating diaspores was counted at 30 min intervals for 180 min from the beginning of the experiment.(c)To test the role of the IO on the floating ability of the diaspores, 50 intact fruits of each of species were weighed using a Sartorius BS210S electronic balance (0.0001 g). Then, the volume of fruits was calculated using the water displacement method [65,66]. A container large enough to hold the fruit was filled with water, and the fruit was carefully sunk in the water until it was completely underwater. For each species, 10 replications of five fruits each were sunk in water, and the volume was calculated as the mean of the five fruits. Then, the displaced water was aspirated with a pipettor (V_p_), and the buoyancy index (*BI*) was calculated following Zeng and Fan using the following equation [65]:

Fruit buoyancy = ρ_w_gV_p_
where ρ_w_ = 10^3^ kg·m^−3^, g = 9.8 N·kg^−1^
BI=FB/FM
where *FB* is the fruit buoyancy and *FM* is the fruit mass.

### 4.6. Statistical Analysis

Before a factor analysis was conducted, data for different diaspores in the wind dispersal treatments were log10 transformed to approximate the normal distribution and homogeneity of variance to fulfill the assumptions of one-way ANOVA. If the ANOVA assumptions were violated after data transformation, treatment differences in these characteristics were assessed by using the more conservative Kruskal–Wallis non-parametric test. One-way ANOVA was conducted to analyze the effect of various types of diaspore traits on dispersal by wind and water. The correlation between phenotypic traits (fruit mass, fruit length, fruit width, fruit thickness, seed mass, seed length, seed width, and seed thickness) and dispersal traits (rate of fall, horizontal dispersal distance, and buoyancy) of the three types of diaspores was analyzed by the Pearson correlation coefficient method. Tukey’s HSD test was used for multiple comparisons to determine if differences between individual treatments were significantly different (*p* < 0.05). Statistics were expressed using mean ± standard error. All data analyses were performed with SPSS version 26.0 (SPSS Inc., Chicago, IL, USA).

## Figures and Tables

**Figure 1 plants-12-01950-f001:**
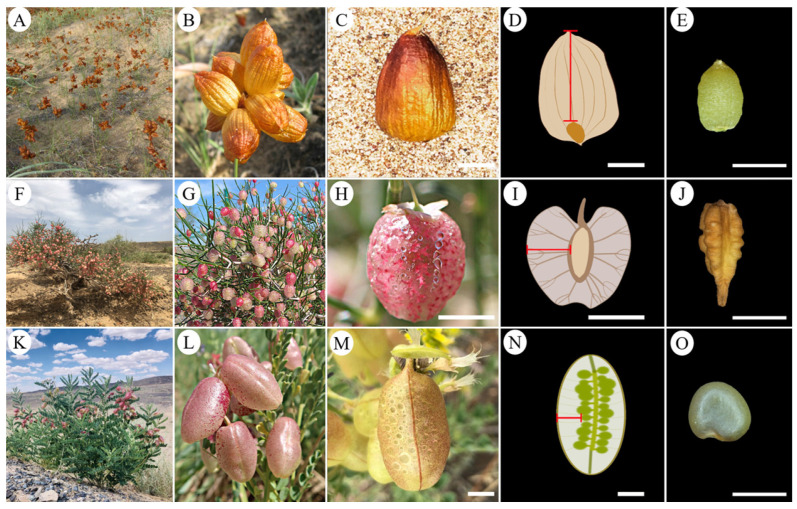
Plant, infructescence, fruit, and seed morphology of the three study species with an inflated ovary in the Karamaili Mountains Nature Reserve, Xinjiang, China. *Carex physodes*, habitat (**A**), infructescence (**B**), fruit (**C**), longitudinal section of IO (**D**), and seed (**E**); *Calligonum junceum*, habitat (**F**), infructescence (**G**), fruit (**H**), longitudinal section of IO (**I**), and seed (**J**); and *Sphaerophysa salsula*, habitat (**K**), infructescence (**L**), fruit (**M**), longitudinal section of IO (**N**), and seed (**O**). Scale bars 5 mm (**C**,**D**,**H**,**I**,**M**,**N**) and 3 mm (**E**,**J**,**O**). Red line represents the distance between the seed and the pericarp.

**Figure 2 plants-12-01950-f002:**
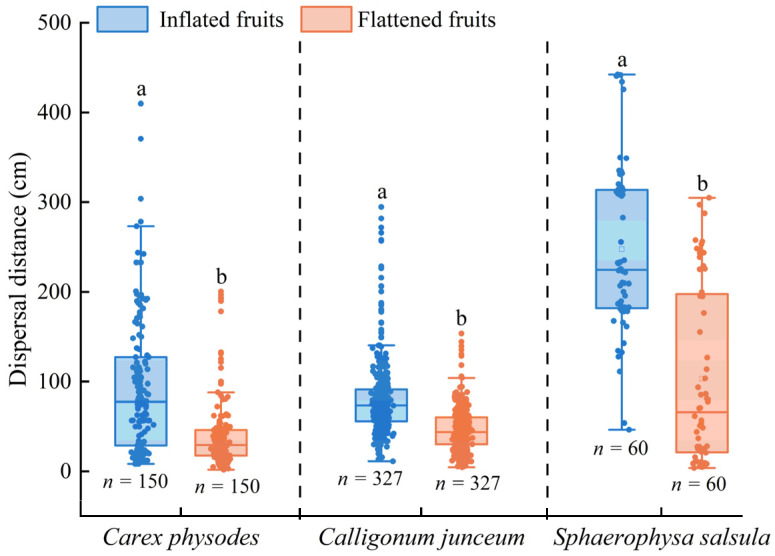
Distribution distance of inflated and flattened fruits of *Carex physodes*, *Calligonum junceum*, and *Sphaerophysa salsula* in the natural habitat. Different lowercase letters indicate significant differences between different treatments of the same species (Tukey’s HSD, *p* < 0.05).

**Figure 3 plants-12-01950-f003:**
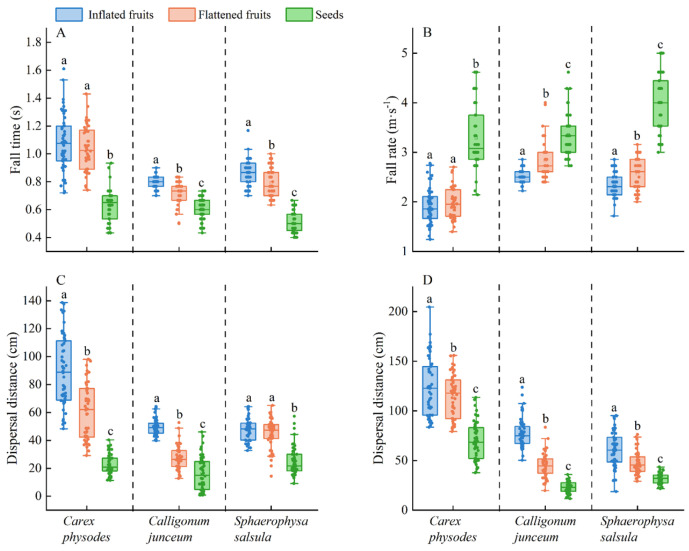
Wind dispersal characteristics of inflated and flattened fruits and seeds of *Carex physodes*, *Calligonum junceum*, and *Sphaerophysa salsula* at two wind speeds: fall time (**A**), fall rate (**B**), and dispersal distance at a wind speed of 4 m·s^−1^ (**C**), and at 6 m·s^−1^ (**D**) (*n* = 100). Different lowercase letters indicate significant differences between different treatments of the same species (Tukey’s HSD, *p* < 0.05).

**Figure 4 plants-12-01950-f004:**
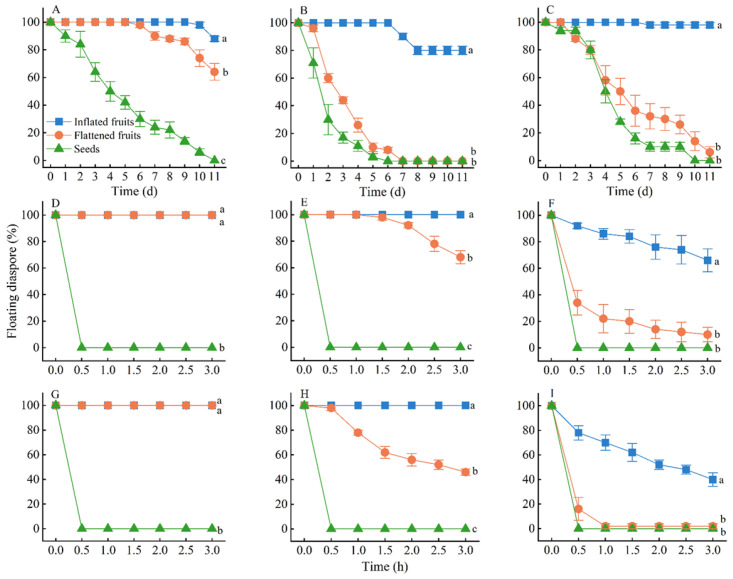
Flotation dynamics of inflated and flattened fruits and seeds of the three study species in still water (**A**–**C**), and water agitated at 80 rpm (**D**–**F**) and 160 rpm (**G**–**I**) in the laboratory. (**A**,**D**,**G**), *Carex physodes*; (**B**,**E**,**H**), *Calligonum junceum*; (**C**,**F**,**I**), *Sphaerophysa salsula* (*n* = 100). Different lowercase letters indicate significant differences between treatments (Tukey’s HSD, *p* < 0.05).

**Figure 5 plants-12-01950-f005:**
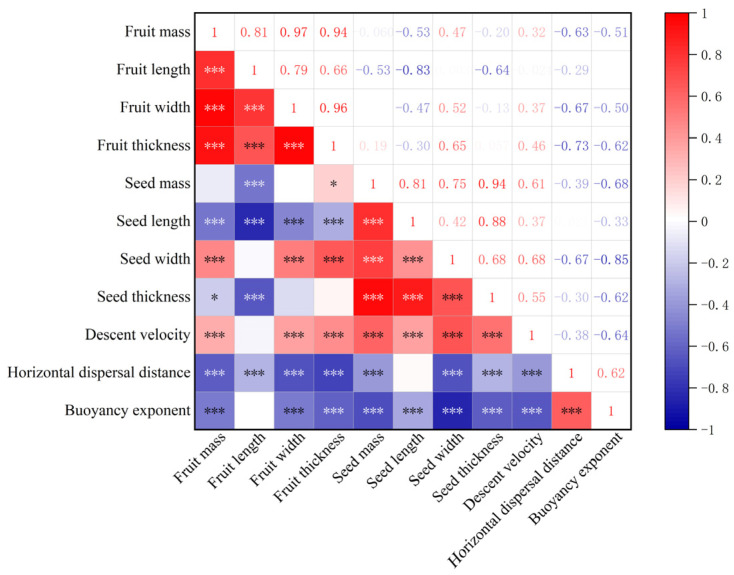
Correlation matrix of inflated ovary and wind dispersal characteristics. Pearson’s correlation coefficient (*r*) between two pairs of variables is shown in the heat map. The correlation coefficients are represented in terms of the changes in the intensity of red/blue color; blue color represents low adjacency (negative correlation), while red represents high adjacency (positive correlation). Statistical significance is indicated by * *p* < 0.05, *** *p* < 0.001.

**Figure 6 plants-12-01950-f006:**
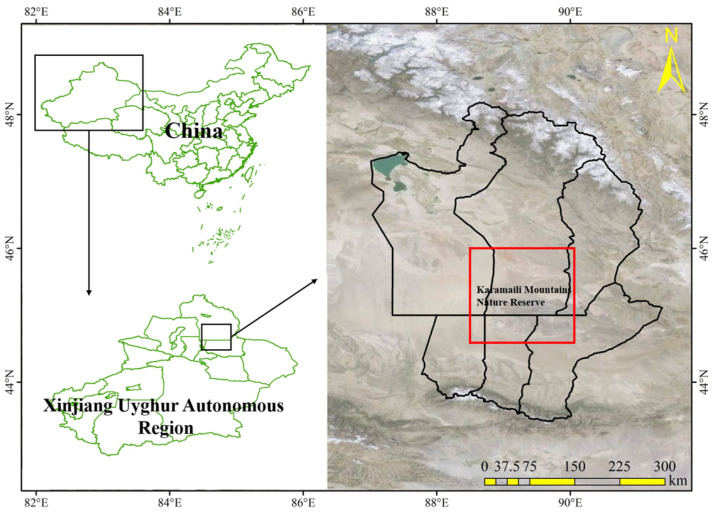
Geographical location of study area (area within a rectangle).

**Table 1 plants-12-01950-t001:** Fruit and seed traits of the three study species (Mean ± SE).

	*Carex physodes*	*Calligonum junceum*	*Sphaerophysa salsula*
Lifestyle	Perennial herbs	Shrubs	Perennial herbs
Habitat	Sandy desert	Gravel desert	Gravel desert
Mass of 100 fruits (g)	0.68 ± 0.01	4.43 ± 0.01	22.24 ± 0.59
Fruit length (mm)	15.16 ± 0.31	9.89 ± 0.85	22.95 ± 0.35
Fruit width (mm)	6.98 ± 0.11	8.86 ± 0.01	18.50 ± 0.21
Fruit thickness (mm)	5.36 ± 0.11	9.86 ± 0.09	15.60 ± 0.23
Buoyancy Index	0.93 ± 0.03	0.25 ± 0.01	0.33 ± 0.01
Mass of 100 seeds (g)	0.23 ± 0.01	1.13 ± 0.02	0.54 ± 0.17
Seed length (mm)	3.39 ± 0.07	4.65 ± 0.05	2.53 ± 0.02
Seed width (mm)	1.69 ± 0.02	2.70 ± 0.04	2.46 ± 0.03
Seed thickness (mm)	0.88 ± 0.01	2.63 ± 0.05	1.14 ± 0.03

## Data Availability

The data that support the findings in the present study are available. from the corresponding author upon request.

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
