# Peer review of "Inflated Ovary May Increase the Dispersal Ability of Three Species in the Cold Deserts of Central Asia"

_plants, 2023, doi:10.3390/plants12101950_

Round 1

Reviewer 1 Report

The article is of high scientific interest. All its sections are congruent among them and well justified.

However, even though the Editorial Norms are quite flexible, it is rare to read "Material and Methods" after "Results" and "Discussion". If "Material and Methods" come first, it is easier to un understand the logic of the "Results" and evaluating if they are well supported by the "Material and Methods" proposed. Hence, I would recommend the "traditional" order: Introduction (including the justification and the objective of the study), Material and Methods, Results, Discussion, and References.

In addition, it is necessary to complete (extended explanation) of the legends of the different Figures and Tables (see file attached), and a map of the study site could be very helpful for the readers, who are not familiar with the Asian geography.

Reviewer 2 Report

Seed and dispersal pathways in Angiosperms is a topic as much invoked in ecological and taxonomic articles as it is dramatically lacking in ad hoc supporting research. For example, increased fitness of seeds included in swollen fruits is often assumed, but rarely proven. For this reason, in my opinion, counter-payments like this are to be encouraged, even if they are apparently easy to conceive and (less obviously) to carry out.

I have a few points that should be clarified. The authors found that the swollen fruits were able to disperse longer in dry, open habitats. However, they found that the intrinsic weight and mass of the seeds also affected distance. Since this weight appears to be related to endosperm content, it might be hasty to relate dispersal distance to seed fitness. Hence, the authors should explicitly state that dispersal distance is obviously not related to germination success. I think germination tests would have been desirable in this case and probably also in the flotation experiments. In fact, we would be sure that a longer stay in water would not affect germinability itself. I understand that this would imply a completely different experimental design, but I advise the authors to discuss this aspect in the ms.

Finally, as a taxonomist, I have serious concerns about the proposed nomenclature. The accuracy of the nomenclature is fundamental in botanical papers, also to allow the replicability of the experiments and to use the interesting results of the work. Fortunately, this gap is more easily filled: please add the authorship of each name mentioned throughout the document and indicate the taxonomic authority you followed (Flora of China?).

As far as my expertise is concerned, I believe that the ms, after adjusting these points, can be published on Plants.
